# Feasibility and Acceptability of Using an Evidence-Based Tai Chi Intervention for Managing the Fatigue–Sleep Disturbance–Depression Symptom Cluster in Breast Cancer Patients

**DOI:** 10.3390/nursrep15050167

**Published:** 2025-05-12

**Authors:** Li-Qun Yao, Tao Wang, Xian-Liang Liu, Jing-Yu (Benjamin) Tan

**Affiliations:** 1College of Nursing, Fujian University of Traditional Chinese Medicine, Fuzhou 350122, China; 2013039@fjtcm.edu.cn; 2Faculty of Health, Charles Darwin University, Brisbane Centre, Brisbane, QLD 4000, Australia; benjamin.tan@unisq.edu.au; 3School of Nursing and Midwifery, University of Southern Queensland, Ipswich, QLD 4305, Australia; 4Centre for Health Research, University of Southern Queensland, Springfield Central, QLD 4300, Australia; 5School of Nursing and Health Sciences, Hong Kong Metropolitan University, Homantin, Kowloon, Hong Kong SAR, China

**Keywords:** Tai Chi, breast neoplasm, symptom cluster, fatigue, sleep disturbance, depression, feasibility, acceptability, randomized controlled trial, qualitative research, nursing

## Abstract

**Objective:** To explore the feasibility and acceptability of using an evidence-based Tai Chi intervention to manage the fatigue–sleep disturbance–depression symptom cluster (FSDSC) in female breast cancer patients. **Methods:** This study reported the feasibility outcomes of a pilot randomized controlled trial (RCT), along with a nested qualitative process evaluation. Seventy-two female breast cancer patients experiencing the FSDSC were randomized into either a Tai Chi group or a control group. The Tai Chi group received an eight-week Tai Chi intervention consisting of two one-hour sessions per week, along with routine care, while the control group received routine care only. The feasibility and acceptability of the study procedure and Tai Chi intervention protocol were assessed by measuring recruitment, referral, retention and drop-out rates, questionnaire completion rates (including the Brief Fatigue Inventory [BFI], Pittsburgh Sleep Quality Index [PSQI], Hospital Anxiety and Depression Scale-Depression [HADS-D], and Functional Assessment of Cancer Therapy-Breast [FACT-B]), intervention adherence, and safety outcomes. The nested qualitative process evaluation consisted of semi-structured interviews conducted among 22 participants to further explore their experiences of participating in this study and practicing Tai Chi. Descriptive data analysis was employed to present the feasibility and acceptability outcomes. Content analysis was employed to analyze the data from the qualitative process evaluation. **Results:** A total of 72 breast cancer patients were successfully recruited over six months, with a recruitment rate of 79.1%, retention rate of 95.8%, and dropout rate of 4.2%. No missing data was found in the BFI, PSQI, or HADS-D. However, a notable number of missing values were found in the FACT-B, particularly for items related to sexual satisfaction. The Tai Chi intervention demonstrated a high level of feasibility, with an average adherence rate of 86.8%. Only eight participants reported minor discomforts, such as minor musculoskeletal discomfort and dizziness, but they were transient and manageable after stopping Tai Chi practice. Semi-structured interviews with 22 participants highlighted that Tai Chi was experienced to be generally convenient, energy-saving, and low intensity for FSDSC management. Participants also felt that the study questionnaires were comprehensible and straightforward. Many interviewees from the Tai Chi group reported perceiving favorable effects on FSDSC management, as well as overall functional health and well-being. **Conclusions:** The evidence-based Tai Chi intervention proved feasible, safe, and convenient as a non-pharmacological intervention for managing FSDSC in breast cancer patients. Future large-scale studies are needed to evaluate Tai Chi’s definite effects on improving FSDSC among breast cancer patients.

## 1. Introduction

Breast cancer poses a significant global health challenge, accounting for a substantial proportion of cancer diagnoses and mortality among women [1]. Breast cancer itself and relevant treatments often give rise to a complex array of symptoms that frequently cluster together [2]. Among breast cancer patients, fatigue, sleep disturbance, and depression have emerged as a particularly prevalent and burdensome symptom cluster [3]. Studies indicated that experiencing symptom clusters can markedly diminish quality of life, impair functional capacity, and complicate social and familial relationships to a greater extent than individual symptoms alone [4,5].

Addressing the complexity and severity of symptom clusters in breast cancer requires the exploration of interventions that are effective, safe, energy-efficient, and low-cost for managing symptoms. Tai Chi, as one of the traditional Chinese exercises (TCEs), has shown promise in alleviating cancer-related individual symptoms, including fatigue, sleep disturbance, and depression [6,7]. However, the available research evidence has not been deemed optimal due to methodological limitations, such as inadequate details on random sequence generation and allocation concealment [8]. Also, to the best of our knowledge, the effects of Tai Chi on symptom cluster management such as the fatigue–sleep disturbance–depression symptom cluster (FSDSC) in cancer patients have not yet been fully explored. Thus, a rigorous investigation into the role of Tai Chi specifically for managing the FSDSC in breast cancer patients is warranted.

However, before launching a full-scale study, a pilot (preliminary) randomized controlled trial (RCT) is essential to evaluate the feasibility and acceptability of a Tai Chi intervention [9]. Thabane et al. emphasize that a pilot study increases the likelihood of success in subsequent main trials and helps avoid the pitfalls of premature full-scale trials [10]. The Medical Research Council (MRC) Framework for Developing and Evaluating Complex Interventions (the MRC framework) highlights the importance of pilot studies to assess both the feasibility of complex interventions and the methodological procedures before launching full-scale RCTs, which are regarded as essential preparatory steps for larger-scale studies [11]. Employing a nested qualitative process evaluation within a clinical trial can provide valuable insights into patients’ perceptions and their experiences of treatment effects [12]. Furthermore, combining qualitative research with complementary and alternative medicine interventional studies in cancer care has been suggested as the most effective way to optimize the intervention programs, as it allows researchers to deeply explore participants’ perspectives, reduce potential biases, and draw more reliable conclusions [13,14].

Our research team, therefore, conducted a research project to develop an evidence-based Tai Chi intervention [15] and assess its feasibility, acceptability, and potential effects on the FSDSC in breast cancer patients through a pilot RCT [9]. This project, grounded in the MRC framework, integrated quantitative feasibility assessments with a qualitative process evaluation, capturing patient experiences and perceptions of Tai Chi practice and study participation to explore the intervention’s feasibility and acceptability [9]. The current paper focuses on the feasibility and acceptability of the Tai Chi intervention and study procedures, presenting key feasibility outcomes of the pilot RCT alongside insights from the qualitative process evaluation, aiming at providing a comprehensive understanding of the intervention’s practicality and the methodological effectiveness of the preliminary RCT.

## 2. Methods

### 2.1. Overview of the Research Project

The overall design of the research project involves two study phases that follow the MRC framework [11,16]. Phase I involved developing and validating an evidence-based Tai Chi intervention on FSDSC management in breast cancer patients, informed by contemporary research, principles and theories of Traditional Chinese Medicine, established practice standards and guidelines, and expert consensus [15]. Phase II comprised a pilot RCT to assess the feasibility and acceptability of the Tai Chi intervention and its potential effects on the FSDSC. Following the pilot RCT, a qualitative process evaluation was performed to explore participants’ experiences with the Tai Chi intervention and participation in the pilot RCT.

The development and validation of the evidence-based Tai Chi intervention [15], the detailed RCT protocol [17], as well as the clinical outcomes of the pilot RCT focusing on the potential effects of Tai Chi on FSDSC [18] have been published. The published pilot RCT demonstrated the promising effects of Tai Chi as an adjunct intervention to routine care in alleviating the symptom cluster of fatigue, sleep disturbances, and depression, while improving the quality of life among breast cancer patients [18]. The current paper focuses on the feasibility and acceptability of the evidence-based Tai Chi intervention and study procedures, presenting key feasibility outcomes of the pilot RCT feasibility outcomes and findings from the nested qualitative process evaluation.

### 2.2. The Pilot RCT Design

This pilot RCT was designed as a two-parallel-arm study. The study took place in two tertiary hospitals in Sichuan and Fujian provinces, China, between May 2020 and May 2021. The protocol was registered prospectively on ClinicalTrials.gov (NCT04190342), and the detailed study protocol was previously published [17]. The RCT design is briefly summarized as follows.

#### 2.2.1. Sample, Randomization, and Allocation Concealment

(1) Study Sample

Adult female patients (aged ≥ 18 years) were eligible for inclusion if they met the following criteria: ① a confirmed diagnosis of stage I–IIIA BC; ② having received adjuvant chemotherapy within the last two months; ③ reported moderate to severe levels of fatigue, sleep disturbance, and depression, as assessed by a 0–10 numerical rating scale (NRS), with scores ranging from 4 to 10 for each symptom in the past month; ④ ability to communicate in Mandarin; ⑤ willingness to provide written informed consent; and ⑥ having undergone BC surgery at least one month prior to enrollment. Exclusion criteria included ① current use of psychostimulants, antidepressants, or hypnotic medications for managing fatigue, sleep disturbance, or depression; ② physical weakness (inability to engage in physical activities due to advanced chronic diseases) or the presence of cognitive impairment and/or severe psychiatric conditions; ③ engagement in other TCE programs exceeding 30 min per session, three times per week, in the last three months; ④ participation in a Tai Chi program in the past six months; and ⑤ elective surgery scheduled during the study period [17,18].

A total sample of 30 participants per group was considered sufficient for this study, given that the primary objective was to evaluate the feasibility and acceptability of the study’s methodological procedures, including the questionnaires and intervention protocol [19,20]. To account for a potential 20% dropout rate, 72 participants (36 per group) were recruited [17,18].

(2) Randomization and allocation concealment

This pilot trial employed a randomized controlled design with a 1:1 allocation ratio, using a computer-generated sequence. To ensure allocation concealment, the randomization sequences were prepared and kept by an independent statistician who had no further involvement in the study. After obtaining informed consent and completing baseline assessment, the two clinical nurses contacted the statistician by telephone to determine group allocation based on the pre-generated sequence. The participants were randomly assigned to either the Tai Chi intervention group or the control group [17,18].

#### 2.2.2. Study Intervention

Participants in both groups received routine care and an educational booklet on cancer symptom self-management, developed from national oncology guidelines, and current research [21,22,23]. A detailed description of the development and validation of the evidence-based Tai Chi protocol is available in a separate methodological publication [15]. Participants in the Tai Chi group underwent an 8-week intervention, consisting of two one-hour sessions per week, which were informed by practice standards/guidelines, theories, current research evidence and experts’ consensus. Each session included a 10 min warm-up, 30 min Tai Chi practice (using an easy 8-form Yang-style Tai Chi), 10 min break, and 10 min cool-down [17,18].

#### 2.2.3. Study Procedure

Detailed procedures were outlined in the published study protocol [17]. Tai Chi group participants attended the standard Tai Chi training sessions to ensure proficiency, with return demonstrations confirming competence. During the subsequent 8-week home practice, participants recorded practice details, and research assistants conducted weekly follow-ups to encourage practice and monitor adverse events [17]. Patient-reported outcomes were collected at baseline, post-intervention, and 4-week follow-up post-intervention using the Brief Fatigue Inventory (BFI, Cronbach’s alpha from 0.90 to 0.92) [24], Pittsburgh Sleep Quality Index (PSQI, Cronbach’s alpha was 0.79) [25], Hospital Anxiety and Depression Scale-Depression (HADS-D, Cronbach’s alpha with a range of 0.82 to 0.90) [26], and Functional Assessment of Cancer Therapy-Breast (FACT-B, Cronbach’s alpha varied between 0.59 and 0.85) [27]. All required permissions were secured, and applicable licensing fees were paid for the use of standardized questionnaires in this study.

#### 2.2.4. Feasibility and Acceptability Outcomes

(1) Recruitment and Follow-up Feasibility

The feasibility and acceptability of the recruitment and follow-up process were assessed using the following measures: (1) time taken to recruit the required number of subjects, (2) referral rate: proportion of referrals from different clinical departments and hospitals relative to all referrals, (3) recruitment rate: proportion of eligible participants enrolled in the study, (4) retention rate: proportion of participants completing the study, (5) dropout rate: proportion of participants withdrawing after randomization, and (6) feedback from participants who discontinued their involvement and identified the reasons for dropout.

(2) Questionnaire Acceptability

Questionnaire acceptability was evaluated by calculating the percentage of missing values for each item and at the scale level in the BFI, PSQI, HADS-D, and FACT-B questionnaires: (a) item-level missing values: percentage of items with no responses or multiple responses where only one was required, and (b) scale-level missing values: percentage of questionnaires with at least one missing item.

(3) Tai Chi Intervention Feasibility and Acceptability

The feasibility and acceptability of the Tai Chi intervention were assessed through (a) adherence rates determined by dividing the number of completed Tai Chi sessions by the total number of scheduled sessions (16 sessions over eight weeks), with adherence rates defined as “good” (>100%), “acceptable” (80–100%), and “poor” (<80%) [28]; (b) participant feedback collected using a researcher-designed questionnaire assessing perceived benefits, satisfaction, and barriers with the Tai Chi program; (c) safety of the intervention: participants reported any discomforts (e.g., musculoskeletal pain, dizziness) during or after Tai Chi sessions, with participants instructed to stop the practice if discomfort occurred; (d) satisfaction with the intervention: participants rated their satisfaction with the Tai Chi intervention on a scale from 1 to 10, where a higher score represents greater satisfaction; and (e) Tai Chi program log: participants in the Tai Chi group maintained a diary recording Tai Chi practice frequency, duration, and any related adverse events after each Tai Chi session.

#### 2.2.5. Data Analysis

Statistical analyses were conducted using IBM SPSS Statistics version 24.0, with a two-sided significance threshold set at 0.05. Comparisons of baseline characteristics between groups were conducted using the Mann–Whitney U test, chi-square test, or Fisher’s exact test as appropriate. Feasibility outcomes (recruitment rates, retention rates, adherence rates, etc.) were presented descriptively including the total percentage, frequency, mean, and standard deviation.

### 2.3. Qualitative Process Evaluation

#### 2.3.1. Sample

Participants were selected for semi-structured interviews from both the control and Tai Chi groups using purposive sampling to ensure diverse perspectives on the therapeutic effects of Tai Chi. Selection criteria involved participants who (a) self-assessed their expectation of Tai Chi effects on a scale from 0 to 10 (0 = “very low expectation” and 10 = “very high expectation”) and (b) indicated whether Tai Chi relieved their FSDSC (yes/no). The sample size for the qualitative process evaluation was determined by data saturation [29].

#### 2.3.2. Procedure

Semi-structured interviews were conducted by a research assistant and the doctoral researcher following the completion of the pilot RCT. After obtaining informed consent to participate in the study, potential interviewees were screened based on the purposive sampling criteria, and interview appointments were scheduled according to their availability. Participants were informed that their involvement in the study was entirely voluntary and that they could withdraw at any time without penalty or consequence. Interviews took place within two months after the Tai Chi intervention via phone calls due to the COVID-19 restrictions. Interviewees were briefed on the interview’s purpose and procedures. A semi-structured interview guide (see Appendix A) was used, which was developed based on previous qualitative and mixed-methods literature on participants’ perceptions of Tai Chi or other TCEs [30,31,32]. The guide was reviewed by academic supervisors who are experienced qualitative researchers for clarity and relevance. Interviews encouraged detailed responses and provided ample time for interviewees to express their perceptions. All interviewees shared their experiences, opinions, and perceptions regarding their participation in the pilot RCT. Interviewees from the Tai Chi group also provided insights specifically on their Tai Chi practice.

#### 2.3.3. Data Analysis

The qualitative data from the interviews were recorded digitally and transcribed by the doctoral researcher, with the transcripts managed in Excel for content analysis. The content analysis process, as outlined by Elo and Kyngäs (2008) [33], comprises three main phases: preparation (selecting the unit of analysis and understanding the data), organizing (open coding, coding sheets, grouping, abstraction, and categorization), and reporting (developing models, conceptual systems, conceptual maps, or categories). To ensure accuracy, transcripts were cross-checked against original recordings by two researchers (the doctoral researcher and one research assistant). Supervisors, research assistants, and the doctoral researcher discussed the data analysis process to maintain rigor and address validity threats. Trustworthiness in qualitative research was obtained through the criteria of credibility, transferability, dependability, and confirmability [34]. Credibility was ensured via triangulation, with two researchers independently coding transcripts and resolving discrepancies with academic supervisors [35]. Transferability was enhanced by providing a detailed description of the characteristics of participants and the qualitative findings of supervisors [35]. Dependability was achieved by thoroughly presenting the study design, data analysis approach, and findings to facilitate external assessment of applicability. Confirmability was maintained through discussions and regular meetings among supervisors, the doctoral researcher, and one research assistant throughout the data analysis of supervisors [35].

### 2.4. Ethical Approval

This research project received ethical approval from the Human Research Ethics Committee at Charles Darwin University (H19094) and the Clinical Trial Ethics Committees at the Affiliated Hospitals of Putian University (201932) and Southwest Medical University (KY2019133). Participants were informed about study objectives, potential benefits and risks, confidentiality measures, and their right to withdraw without consequences. All participants provided written informed consent before their involvement in the whole study.

## 3. Results

### 3.1. Participants’ Baseline Information

The mean age of participants was 46.9 years old. The majority had a high school education or lower (62, 86.1%) and were married (68, 94.4%). Most participants had been diagnosed with stage II or IIIA breast cancer (50, 69.4% and 13, 18.1%, respectively), with over half (41, 56.9%) having undergone six chemotherapy cycles. Baseline demographic and clinical characteristics did not significantly differ between the Tai Chi and control groups post-randomization (*p* > 0.05). More details were presented in the clinical outcome paper [18].

### 3.2. Feasibility and Acceptability Results

#### 3.2.1. Feasibility of Recruitment and Follow-Up

Over the six-month recruitment period (May to November 2020), 126 breast cancer patients were screened, leading to the enrollment of 72 participants. Referrals from Oncology Department clinicians accounted for only two participants. Comparable recruitment rates were observed between the two study sites, with 48.6% (35/72) from the Affiliated Hospital of Putian University and 51.4% (37/72) from the Affiliated Hospital of Southwest Medical University (*p* = 0.099). The eligibility and recruitment rates were 72.2% (91/126), with 35 not meeting the inclusion criteria, and 79.1% (72/91), with 19 declining to participate, respectively. Of the 72 participants, 95.8% (69/72) completed the trial, reflecting a low dropout rate of 4.2% (3/72), with all three dropouts from the control group [18]. Reasons for dropout included loss of contact and hospital transfer due to lung metastasis.

#### 3.2.2. Acceptability of Questionnaires

The study questionnaires (BFI, PSQI, HADS-D, FACT-B) were found to be highly acceptable, with no missing data in the BFI, PSQI, or HADS-D questionnaires at all three time points. However, the FACT-B questionnaire exhibited a notable number of missing values, particularly for the item related to sexual satisfaction (FACT-B-GS 7). At baseline, 23.6% of participants refused to answer item FACT-B-GS 7, while missing values for the other four items (item FACT-B-GP3, item FACT-B-GS3, item FACT-BGS6, and item FACT-B-B4) were minimal (all at 1/72, 1.40%). Post-intervention and follow-up assessments showed a significant refusal rate for item FACT-B-GS 7, with 52.9% and 55.1% of participants, respectively, declining to answer. The completion rates for the entire FACT-B questionnaire ranged from 43.5% to 76.4%. The missing values for all four questionnaires at the scale level at three time points are detailed in Table 1.

#### 3.2.3. Adherence and Acceptability to the Tai Chi Intervention

(1) Total Number of Tai Chi Intervention Weeks

Adherence regarding the Tai Chi intervention weeks is outlined in Table 2. Among the 36 participants in the Tai Chi group, 69.4% (25/36) successfully completed the eight-week program, with 41.7% (15/36) adhering strictly to the protocol. The average adherence rate was 86.8%, with 80.5% (29/36) demonstrating good or acceptable adherence.

(2) Frequency and Duration of Tai Chi Intervention

Regarding session frequency, more than half of the participants (20/36, 55.6%) adhered to the prescribed regimen of twice-weekly Tai Chi practice. Concerning session duration, 16 participants (44.4%) adhered to the recommended one-hour sessions, while 15 participants (41.7%) engaged in sessions of shorter duration. Over the eight-week intervention period, 25 participants (69.4%) completed at least 16 Tai Chi sessions. More than half of the participants (20/36, 55.6%) accumulated a total Tai Chi practice time of at least 16 h throughout the intervention (Table 3).

(3) Participants’ Feedback and Satisfaction

Regarding participants’ feedback (Table 4), significant psychological and physical benefits from Tai Chi practice were reported, with mean scores ranging from 3.7 to 4.3 on a 1–5 scale, where higher values indicate greater benefits. Over 90% of participants noted Tai Chi’s effectiveness in alleviating emotional distress (e.g., anxiety and depression), fostering a relaxed spiritual state, and reducing life pressures. Additionally, approximately 70% reported Tai Chi’s beneficial effects for enhancing attention, while 91.7% experienced improved joint and muscle relaxation. Participants also reported enhancements in vitality, sleep quality, and bodily flexibility. Furthermore, a significant number of participants indicated increased sociability and energy levels post-intervention. Lack of time to practice and having difficulties in managing the techniques were occasionally reported as the challenges in practicing Tai Chi, with percentages of 30.5% and 11.1%, respectively. Participants generally expressed high levels of satisfaction with the Tai Chi intervention, with mean scores of 8.4 for overall satisfaction.

(4) Safety of the Tai Chi Intervention

Eight participants noted minor adverse effects. These included five cases of mild musculoskeletal discomfort occurring either during or after Tai Chi sessions, two cases of minor dizziness arising from a rapid practice pace, and one participant reporting minor knee pain. Participants indicated that these discomforts were transient and tolerable, subsiding shortly after discontinuation of the practice and subsequent rest.

### 3.3. Findings from the Qualitative Process Evaluation

#### 3.3.1. Participants’ Demographic Data

Twenty-two participants (15 from the Tai Chi group, 7 from the control group) were included in the qualitative process evaluation, with a mean age of 42.86 years old. Further details on the characteristics of the interviewees are presented in Table 5.

#### 3.3.2. Categories

Four main categories emerged from qualitative data, capturing participants’ experiences and perceptions of the pilot RCT and Tai Chi intervention, with two to five subcategories under each category. The details of the main categories, subcategories, and quotes are presented in Appendix B.

Category 1: General Perspectives on TCEs

The majority of the participants in the study regarded TCEs, such as Tai Chi, Qigong, and Baduanjin, as widely accepted and easily accessible forms of physical activity. For example, one participant stated, “*Um…I [think TCEs are] pretty good, interesting and convenient. Many people surrounding are practicing TCEs, particularly practicing in the park. TCEs are quite popular and convenient and can be practiced everywhere.*” Many participants acknowledged that their acquaintances practiced TCEs. Participants believed that TCEs can enhance their overall well-being and support rehabilitation, particularly through their capacity to regulate bodily functions during recovery from illness. They held high expectations regarding the potential of TCEs to alleviate cancer-related symptoms, including fatigue, sleep disturbances, and depression. This optimism stemmed from the gentle, coordinated nature of TCE movements.

Category 2: Experiences with Tai Chi Practice during the Pilot RCT

Most participants in the Tai Chi group reported symptom management benefits and supported its integration into routine cancer care. They emphasized its convenience, safety, and low-intensity nature, which are particularly beneficial for individuals with fatigue. One participant highlighted, “*Yes, I will continue to practice Tai Chi, I practiced Tai Chi all the time, because practicing Tai Chi is a more convenient intervention compared with other exercises and can be practiced anywhere and anytime. For example, if it was raining or cold outside, I could practice at home. Previously, I tried jogging to improve my well-being, but I found I could not run anymore after 200 to 300 m, it made me too tired to run. Um…also, after jogging I felt my heart rate was too fast, breathless, so I think jogging maybe too [high] intensity for me.*” Participants generally adhered to the Tai Chi protocol of twice-weekly, hour-long sessions, despite challenges such as busy household schedules, limited motivation for independent practice, lack of confidence in their ability to practice well, and fatigue. Furthermore, some participants preferred group practice, highlighting the benefits of peer support. They perceived group sessions as fostering positive social interactions, mutual encouragement, and emotional support, which could bolster adherence and sustained engagement in Tai Chi practice. Many participants appreciated the follow-up calls as a caring approach, providing them with emotional support and valuable information.

Category 3: Experiences with Data Collection in the Pilot RCT

Participants generally found that all the study questionnaires were straightforward, easy to understand, and simple to complete. Most agreed that the time required (10–15 min) was reasonable. They also felt that these tools effectively captured and described their complex experiences related to cancer symptoms. Some participants mentioned gaining valuable knowledge about their symptoms and management strategies through the questionnaires. For example, one interviewee stated, “*I found knowledge from the PSQI that there are many reasons that can contribute to sleep disturbance, such as hot flashes or waking early.*” While there were minor complaints about the number of questions, only two participants reported that the number of items was burdensome and time-consuming, particularly given their fatigue from chemotherapy.

Category 4: Perceptions of Tai Chi Effects

More than half of the participants in the Tai Chi group reported significant relief from fatigue, sleep disturbances, and depression following the eight-week Tai Chi intervention. Most noted relief in at least one of these symptoms, attributing the improvement directly to Tai Chi practice. Participants perceived Tai Chi as therapeutic, noting that their symptoms worsened when the Tai Chi practice ceased, which reinforced their belief in its efficacy. For example, the interview stated, “*I believe Tai chi can effectively relieve fatigue, sleep disturbance, and depression, particularly for fatigue and sleep problems. After taking some physical exercise, I could actually sleep better. If I didn’t practice Tai chi in the day, I would wake up once or twice during the night, and my sleep quality would not be good. [The benefits of Tai chi were] making you more relaxed, and you can also feel…and it also improved my emotional status. When you practice Tai chi in the park with the fragrance of the flowers and the fresh air, you feel indeed pleasant.*” Furthermore, many participants mentioned improved overall functional status, both physical and psychological, after the eight-week program. They reported enhancements in strength, dexterity, and endurance, comparing their functional abilities before and after the intervention. Additionally, some participants described experiencing a sense of peace and tranquility through regular Tai Chi practice.

## 4. Discussion

The MRC framework emphasizes the value of integrating quantitative and qualitative approaches when evaluating complex interventions [11]. The primary objective of this pilot RCT was to examine the feasibility of implementing an evidence-based Tai Chi program for managing the FSDSC in BC patients. In addition to quantitative outcomes, qualitative feedback provided critical insights into participants’ experiences with the RCT procedures and the overall acceptability of the study protocol. These findings are essential for refining the design and methodology of a future full-scale RCT aimed at rigorously evaluating the effects of Tai Chi in managing FSDSC among BC patients.

### 4.1. Feasibility of Recruitment and Follow-Up Process

The study successfully achieved its objectives of testing recruitment procedures and feasibility. The face-to-face recruitment strategy proved effective, resulting in a high recruitment rate even during the COVID-19 period. The high level of recruitment and retention in this study indicated enthusiasm among the breast cancer patients and their interest in Tai Chi. The recruitment rate in this study not only surpasses benchmarks from similar studies on exercise interventions for cancer-related symptoms [36,37] but also exceed the median recruitment rate of 49% noted in a systematic review [38]. High recruitment in this study may be attributed to several factors: Tai Chi’s popularity, convenience, and safety. This was further supported by the qualitative process evaluation, in which most interviewees expressed positive attitudes toward Tai Chi and reported high expectations regarding the potential benefits of traditional Chinese exercises (TCEs) for alleviating cancer-related symptoms. Additionally, the strong rapport established between oncology nurses and patients through routine clinical care likely further enhanced recruitment and engagement in the study.

In terms of retention, the study achieved an excellent rate, with only three participants withdrawing, surpassing the average retention rate of 86% reported in a systematic review of exercise interventions among cancer patients [39]. This success may be attributed to factors including the mentioned convenience of Tai Chi, accommodating participant schedules without undue burden. Weekly phone reminders played a vital role in maintaining retention by reinforcing participant dedication and providing emotional support, as evidenced by qualitative insights from the study, where participants highlighted the positive impact of these reminders on their engagement. Additionally, the early-stage breast cancer status likely contributed to the high retention rates, given their milder symptom burden and higher energy levels compared to those with advanced cancer stages [40]. All withdrawals occurred within the control group, suggesting reasons unrelated to the Tai Chi intervention and reflecting the high acceptability of Tai Chi intervention among the participants.

### 4.2. Acceptability of Assessments

The quantitative data showed that the absence of missing values in BFI, PSQI, and HADS-D items indicated their suitability for the study population, supported by the simplicity of their numeric rating systems [41,42,43]. The early-stage cancer patient cohort likely contributed to high completion rates due to their physical capability and motivation to engage with the questionnaires, reflecting satisfaction with the tools [44]. This aligns with qualitative feedback from the majority of participants affirming that the questionnaires were manageable within their daily routines, where participants consistently described the questionnaires as “clear”, “informative”, and “easy to complete”. However, high non-response rates to the sexual satisfaction item in FACT-B (FACT-B-GS7: up to 55.1% post-intervention) reflect culturally sensitive discomfort. Despite the challenges in FACT-B related to sexuality items, the overall instrument acceptability was maintained through specialized scoring techniques.

### 4.3. Adherence and Acceptability to the Tai Chi Intervention Protocol

The feasibility of the Tai Chi intervention was supported by high adherence and positive feedback. Adherence rates in this study exceeded those of similar exercise interventions (e.g., Qigong, dancing) for breast cancer patients, ranging from 60% to 82% [5,45,46]. Key factors related to this robust adherence may include perceived significant physical and psychological benefits, cultural belief in Tai Chi’s overall health benefits among the Chinese population, and the flexible design of the intervention. These assumptions were supported by the findings of the qualitative process evaluation, which highlighted significant improvements in fatigue, sleep disturbances, and depression, suggesting perceived benefits of Tai Chi in managing symptoms among breast cancer patients.

Despite the favorable responses, several participants faced challenges such as time constraints and difficulties in mastering Tai Chi techniques, which were consistent with previous research on mind–body exercises [38,47]. Specifically, Chinese women reported a lack of time due to caregiving duties, reflecting Asian cultural values emphasizing filial piety [48]. Occasional memory issues also posed challenges during practice sessions. Recommendations included implementing dynamic skill assessments via digital platforms (e.g., WeChat) to support participants with memory challenges and optimize practice effectiveness [49]. Overall satisfaction with the Tai Chi intervention was high, with participants expressing willingness to continue and recommend Tai Chi to others facing similar health challenges.

Safety evaluations indicated no serious adverse events related to Tai Chi practice throughout the study. The intervention’s low-intensity nature and tailored protocol likely contributed to its safety profile, supported by participant feedback indicating Tai Chi as a safe and low-intensity intervention. Minor reactions, such as musculoskeletal discomfort, were reported early in the intervention but resolved without medical intervention upon discontinuation of practice. These occurrences may be attributed to initial adaptation to Tai Chi movements and improper technique execution, emphasizing the importance of adequate instruction and supervision [50].

## 5. Refinement and Implications for Future Research

This pilot study feasibility assessment provides valuable insights for refining the protocol of a future large-scale RCT. Although the home-based Tai Chi format was convenient, some participants preferred group sessions for peer support and enhanced compliance. Future interventions can consider incorporating group-based Tai Chi programs led by health professionals or peers to provide enhanced emotional and social support. To address difficulties in mastering Tai Chi techniques and issues related to inappropriate practice, dynamic assessments can be introduced. Bi-weekly face-to-face skills assessments can be considered to ensure proper technique and address any challenges, thereby improving safety and confidence in the practice. Lastly, given the traditional conservative belief that Chinese women often refuse to discuss sensitive topics openly, particularly their sex life [51,52] strategies to enhance privacy during sensitive assessments can be implemented to mitigate cultural barriers and improve data completeness.

## 6. Conclusions

Findings from the preliminary RCT and the nested qualitative process evaluation demonstrated that the Tai Chi intervention and RCT methodological procedures were feasible for FSDSC management in breast cancer patients, with promising results for recruitment, retention, assessment acceptability, and the acceptability of Tai Chi interventions. These findings support a further large-scale RCT to validate Tai Chi as a safe and feasible intervention in breast cancer care.

## Figures and Tables

**Table 1 nursrep-15-00167-t001:** Missing values for all four questionnaires at three time points.

Time Point	Questionnaires	BFI	HADS-D	PSQI	FACT-B
Baseline assessment (n = 72)	No. of participants who answered all the items	72	72	72	55
	No. of participants with missed item (%)	0 (0%)	0 (0%)	0 (0%)	17 (23.6%)
Post-intervention assessment (n = 70) ∆	No. of participants who answered all the items	70	70	70	33
	No. of participants with missed item (%)	0 (0%)	0 (0%)	0 (0%)	37 (52.9%)
Follow-up assessment (n = 69) ∆	No. of participants who answered all the items	67	68	68	30
	No. of participants with missed item (%)	0 (0%)	0 (0%)	0 (0%)	39 (56.5%)

Note: BFI: Brief Fatigue Inventory; PSQI = Pittsburgh Sleep Quality Index; HADS-D = Hospital Anxiety and Depression Scale-Depression; FACT-B = Functional Assessment of Cancer Therapy-Breast. ∆ = Two participants dropped out during the post-intervention assessment, and one participant dropped out during the four-week follow-up assessment. The number of subjects for post-intervention and follow-up assessments was 70 and 69, respectively.

**Table 2 nursrep-15-00167-t002:** Total number of Tai Chi intervention weeks in the Tai Chi group.

Tai Chi Intervention Weeks	Tai Chi Group (n = 36), Number (%)
8-week Tai Chi intervention	25 (69.4%)
7-week Tai Chi intervention	2 (8.3%)
6-week Tai Chi intervention	2 (5.6%)
5-week Tai Chi intervention	2 (5.6%)
4-week Tai Chi intervention	3 (5.6%)
3-week Tai Chi intervention	2 (5.6%)
2-week Tai Chi intervention	0 (0%)
1-week Tai Chi intervention	0 (0%)
Strictly Adhered to Tai Chi Protocol	
Yes (meeting the required dosage: 8 weeks of Tai Chi intervention, 2 sessions per week, 1 hour per session)	15 (41.7%)
No (not meeting the required dosage)	21 (58.3%)
Adherence Rate *	86.8%
Good adherence (>100%)	8 (22.2%)
Acceptable adherence (80% to 100%)	21 (58.3%)
Poor adherence (<80%)	7 (19.4%)

Note: * = adherence rate: the number of Tai Chi sessions completed divided by the total scheduled sessions (16 sessions over 8 weeks); the adherence rates were categorized as “good adherence”, “acceptable adherence”, and “poor adherence”, according to the criteria established in the study by Huang et al. [28].

**Table 3 nursrep-15-00167-t003:** Frequency and duration of the Tai Chi intervention in the Tai Chi group.

Adherence to Frequency and Duration of Practicing Tai Chi	Tai Chi Group (n = 36) Number (%)
Frequency of Tai Chi	
<Twice per week	4 (11.1%)
Twice per week (standard)	20 (55.6%)
>Twice per week	12 (33.3%)
Duration of Tai Chi	
≤0.5 h each time	2 (5.6%)
>0.5 h < 1 h each time	13 (36.1%)
1 h each time (standard)	16 (44.4%)
>1 h each time	5 (13.9%)
Total Frequency of Tai Chi During 8-Week Tai Chi Intervention	
<16 times	11 (30.6%)
16 times (standard)	17 (47.2%)
>16 times	8 (22.2%)
Total Duration of Tai Chi During 8-Week Tai Chi Intervention	
≤8 h	7 (19.4%)
>8 h < 16 h	9 (25.0%)
16 h (standard)	15 (41.7%)
>16 h	5 (13.9%)

**Table 4 nursrep-15-00167-t004:** Summary of participants’ feedback and satisfaction with the Tai Chi intervention.

Participants’ Feedback	Tai Chi Group (n = 36) Number (%)					
	Strongly Agree	Agree	No Opinion	Disagree	Strongly Disagree	∆ Mean/SD
Emotional relief	11 (30.6%)	23 (63.9%)	2 (5.6%)	0 (0%)	0 (0%)	4.3/0.6
Attention enhancement	8 (22.2%)	17 (47.2%)	11 (30.6%)	0 (0%)	0 (0%)	3.9/0.7
Life pressure relief	9 (25.0%)	19 (52.8%)	8 (22.2%)	0 (0%)	0 (0%)	4.0/0.1
Anxiety relief	10 (27.8%)	23 (63.9%)	3 (8.3%)	0 (0%)	0 (0%)	4.2/0.1
Depression relief	9 (25.0%)	24 (66.7%)	3 (8.3%)	0 (0%)	0 (0%)	4.2/0.1
Joint and muscle relaxation improvement	10 (27.8%)	23 (63.9%)	3 (8.3%)	0 (0%)	0 (0%)	4.2/0.1
Joint and muscle flexibility improvement	8 (22.2%)	17 (47.2%)	11 (30.6%)	0 (0%)	0 (0%)	3.9/0.1
Relaxed spiritual mood	11 (30.6%)	22 (61.1%)	3 (8.3%)	0 (0%)	0 (0%)	4.2/0.1
Vitality improvement and tiredness relief	10 (27.8%)	22 (61.1%)	4 (11.1%)	0 (0%)	0 (0%)	4.2/0.1
Feel relaxed and energetic	7 (19.4%)	15 (41.7%)	14 (38.9%)	0 (0%)	0 (0%)	3.8/0.1
Sociability enhancement	6 (16.7%)	13 (36.1%)	17 (47.2%)	0 (0%)	0 (0%)	3.7/0.1
Quality of sleep improvement (e.g., fall asleep easily)	7 (19.4%)	24 (66.7%)	5 (13.9%)	0 (0%)	0 (0%)	4.1/0.1
Difficulties in Practicing Tai Chi	**Never**		**Occasionally**		**Often**	**No Opinion**
Lack of time to practice at home	23 (63.9%)		11 (30.5%)		2 (5.6%)	0 (0%)
Difficulties in managing the techniques	32 (88.9%)		4 (11.1%)		0 (0%)	0 (0%)
Noisy home environment	34 (94.4%)		2 (5.6%)		0 (0%)	0 (0%)
No interest in Tai Chi	35 (87.2%)		0 (0%)		1 (2.8%)	0 (0%)
Variables			**Tai Chi Group (n = 36)**			
			**^#^ Mean/SD/SE/Median [Range]**			
Satisfaction with Tai Chi intervention (1–10)			8.4/1.1/0.2/9.0 [5–10]			
Consideration of further Tai Chi practice (1–10)			8.5/1.1/0.2/9.0 [5–10]			
Willingness to recommend Tai Chi to others (1–10)			8.8/1.0/0.2/9.0 [5–10]			

Note: ∆ **=** rating was made using a 5-point Likert scale (5 = strongly agree, 4 = agree, 3 = no opinion, 2 = disagree, 1 = strongly disagree); ^#^ = rating was made using a 10-point Likert scale, with 10 indicating either “very dissatisfied” or “absolutely yes”; SD = standard deviation; SE = standard error.

**Table 5 nursrep-15-00167-t005:** Summary of the interviewees’ characteristics.

Variables		Number (%)
Group allocation of the pilot RCT (n = 22)	Tai Chi group	15 (68.2%)
	Control group	7 (31.8%)
Age (n = 22)	Maximum age	55
	Minimum age	27
	Mean	42.86
Education level (n = 22)	Primary school	5 (22.7%)
	Secondary school	6 (27.3%)
	High school/technical school	5 (22.7%)
	College diploma/university degree or above	6 (27.3%)
Marital status (n = 22)	Married	21 (95.5%)
	Single	1 (4.5%)
Employment status (n = 22)	Professional	4 (18.2%)
	Housewife	3 (13.6%)
	Administrative/clerical	3 (13.6%)
	Retired/unemployed	12 (54.5%)
Breast cancer stage (n = 22)	I	5 (22.7%)
	IIA	7 (31.8%)
	IIB	4 (18.2%)
	IIIA	6 (27.3%)
Chemotherapy combination (n = 22)	AC/ACT combination	9 (40.9%)
	EC/EC-T/EC-D combination	10 (45.5%)
	TC combination	3 (13.6%)
Previous experience with TCEs (n = 22)	Yes	5 (22.7%)
	No	17 (77.3%)
Adherence to Tai Chi protocol (n = 15)	Rigid adherence **^#^**	8 (53.3%)
	No rigid adherence	7 (46.7%)
Expectations of the Tai Chi intervention effects * (n = 15)	Maximum score	10
	Minimum score	7
	Mean score	8.60

Note: RCT = randomized controlled trial; AC = doxorubicin and cyclophosphamide; EC = epirubicin and cyclophosphamide; T = paclitaxel; TC = cyclophosphamide and docetaxel; D = docetaxel; TCEs = traditional Chinese exercises; ^#^ = participants strictly adhered to the researchers’ instructions to practice Tai Chi twice a week, with each session lasting one hour; * = expectations of the Tai Chi intervention effects were assessed using a 0 to 10 numerical scale (0 = “very low expectation” and 10 = “very high expectation”).

## Data Availability

The data supporting the findings of this study are included within the article. Additional data may be available upon reasonable request to the corresponding author, subject to applicable restrictions.

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
