# Peer review of "Feasibility and Acceptability of Using an Evidence-Based Tai Chi Intervention for Managing the Fatigue–Sleep Disturbance–Depression Symptom Cluster in Breast Cancer Patients"

_nursrep, 2025, doi:10.3390/nursrep15050167_

Round 1
Reviewer 1 Report
Comments and Suggestions for Authors
Comments and suggestions for authors.
Cancer and its effects, both somatic and psychological, leave their many side effects affecting functioning. Tai chi is an ancient Chinese art not only of combat, but also of meditation and relaxation. Exercise in slow, fluid movements combined with breathing techniques have a relaxing and restorative effect. The authors undertook a study of the effects of Tai chi technique on patients undergoing oncological treatment for breast cancer and the effects of the disease on the mental sphere. According to the reviewer, several corrections and clarifications should be made:
- the symptoms associated with cancer have been described in many articles, not as the researchers write verse 80 correct;
- the description of the research project should be supplemented with clear principles rather than pointing the reader to sources;
- lack of refined inclusion and exclusion criteria for the study;
- the frequency and duration of the Tai Chi intervention on what factors did it depend;
- Whether participants were allowed to discontinue the study at any stage;
- Whether the researchers obtained consent to use standardized questionnaires;
- cronbach's α coefficient value.
- How was the correlation performed? How was the data summarized for correlation analysis considering multiple categories/variables? To clarify whether strongly agree and agree on both statements were summarized as one?
- to what extent did the indisposition in the study e.g. lack of time, difficulty in mastering the exercises limit the impact on the result?
- The discussion lacks specifics from the study and mostly links to other studies;
Reviewer 2 Report
Comments and Suggestions for Authors
This manuscript is an RCT based on breast cancer survivors using Tai Chi practice to alleviate fatigue sleep depression cycle.
- Please add numbers of descriptive statistics in the form of N (%) for 'Participant's Baseline Information.
- Although saturation method was used for qualitative analysis, its results are not clearly mentioned. Need to write what common themes or patterns were found by qualitative analysis. Use double quotations to show the verbatim quotes from the participants.
- Category 3: Experience with Data Collection does not serve any purpose in the results section of the paper. The validation of the data collection methods is something that the researchers should do at the backend, and those results need not be included in the paper. Should delete this entire section.
- Category 4: Perceptions of Tai Chi Effect. Authors should include some quotes from the participants and identify some common themes among participants in relation to their perception of the benefits of Tai Chi in relieving FSDSC.
- Consider adding a conceptual diagram based on a behavioral theory which fits the results from your qualitative analysis. Show a figure which demonstrates the interplay of different factors which Tai Chi exercise benefit in these cancer survivors.
- Discussion section should talk about how the qualitative and quantitate results support the study hypothesis. What was similar and different in terms of results from each of these analyses.
- Line 420 to 423 are two sentences saying the same thing essentially, it sounds repetitive, please revise it.
- Not sure what is meant by 'Acceptability of Assessments', did the participants have an option of not accepting the assessments. Was this acceptability rated by the participants.
- 4.2 sounds similar to category 3 from the results section, not sure the purpose it serves, other than the participants feedback on the different questionnaires, which should have been something that can be included in one sentence. The results do not provide any significant information.
Reviewer 3 Report
Comments and Suggestions for Authors
It was a pleasure to read the article! Good structure, clear and easy reading, interesting topic and results! Hope for a extend study and overall positive benefits! The cluster symptoms and assessments are of a real interest in daily practice.
The only comments are:
- The date/period of the study should be included in the methodology.
- Table 5
Please explain how “Rigid adherence” was defined
Reviewer 4 Report
Comments and Suggestions for Authors
This an interesting research paper, outlining in the introduction section the research question/aim. Therefore, the authors attempted to evaluate the value of traditional tai-chi exercise as part of improving quality of life for cancer patients. They implemented a well designed research protocol, where patients accepted a tai-chi intervention program and they were compared to a control group of patients that did not follow the program. At the end the results were documented on a properly designed and scientifically accurate questionnaire. This is a pilot study, that as the authors state will be used for a larger scale study, in the future.
The results are well presented and the authors adequately state the limitations of their study in the discussion session. Nevertheless, it seems that there is one point in the method design that needs better explanation. The authors should elaborate more in explaining the method they used for deciding on the number of participants in order to achieve a good power of analysis.
Round 2
Reviewer 1 Report
Comments and Suggestions for Authors
The authors addressed the reviewer's guidance in their article. It does not raise objections in terms of content and methodology.